



# A new global grid-based weighted mean temperature model considering vertical nonlinear variation

Peng Sun[1], Suqin Wu[1,2], Kefei Zhang[1,2], Moufeng Wan[1], Ren Wang[1]

[1] School of Environment Science and Spatial Informatics, China University of Mining and Technology, Xuzhou 221116, China
[2] SPACE Research Center, School of Science, RMIT University, Melbourne 3001, Australia

*Correspondence to*: Suqin Wu (sue.wu2018@gmail.com)

**Abstract.** Global Navigation Satellite Systems (GNSS) have been proved to be an excellent technology for retrieving precipitable water vapor (PWV). In GNSS meteorology, PWV at a station is obtained from a conversion of the zenith wet delay (ZWD) of GNSS signals received at the station using a conversion factor which is a function of weighted mean temperature ($T_m$) along the vertical direction in the atmosphere over the site. Thus, the accuracy of $T_m$ directly affects the quality of the GNSS-derived PWV. Currently, the $T_m$ value at a target height level is commonly modelled using the $T_m$ value at a specific height and a simple linear decay function, whilst the vertical nonlinear variation in $T_m$ is neglected. This may result in large errors in the $T_m$ result for the target height level, as the variation trend in the vertical direction of $T_m$ may not be linear. In this research, a new global grid-based $T_m$ empirical model with a horizontal resolution of 1°× 1°, named GGNTm, was constructed using ECMWF ERA5 monthly mean reanalysis data over the 10-year period from 2008 to 2017. A three-order polynomial function was utilized to fit the vertical nonlinear variation in $T_m$ at the grid points, and the temporal variation in each of the four coefficients in the $T_m$ fitting function was also modelled with the variables of the mean, annual and semi-annual amplitudes of the 10-year time series coefficients. The performance of the new model was evaluated using its predicted $T_m$ values in 2018 to compare with the following two references in the same year 1) $T_m$ from ERA5 hourly reanalysis with the horizontal resolution of 5°× 5°; 2) $T_m$ from atmospheric profiles from 428 globally distributed radiosonde stations. Compared to the first reference, the mean RMSEs of the model predicted $T_m$ values over all global grid points at the 950hPa and 500hPa pressure levels were 3.35K and 3.94K respectively. Compared to the second reference, the mean bias and mean RMSE of the model predicted $T_m$ values over the 428 radiosonde stations at the surface level were 0.34K and 3.89K respectively; the mean bias and mean RMSE of the model's $T_m$ values over all pressure levels in the height range from the surface to 10 km altitude were −0.16K and 4.20K respectively. The new model results were also compared with that of the GPT3, GTrop and GWMT_D models in which different height correction methods were also applied. Results indicated that significant improvements made by the new model were at high-altitude pressure levels; in all five height ranges, GGNTm results were generally unbiased, and their accuracy varied little with height. The impact of $T_m$ on GNSS-PWV was evaluated in terms of relative error, and significant improvement was found compared to the widely used GPT3 model. These results suggest that considering the vertical nonlinear variation in $T_m$ and the temporal variation in the coefficients of the $T_m$ model can significantly improve the accuracy of model-predicted Tm for a GNSS receiver that is located in anywhere below the



tropopause (assumed to be 10 km), which has significance for applications needing real-time or near real-time PWV converted from GNSS signals.

## 1 Introduction

Water vapor, as an important greenhouse gas, is tightly related to weather variations, hence, it is crucial to monitor the water vapor content in the atmosphere for reliable weather forecast. The meteorological parameter that is closely related to water vapor is precipitable water vapor (PWV) and it can be measured by various technologies such as radiosondes, remote sensing satellites and water vapor radiometers. Global Navigation Satellite Systems (GNSS), which were initially designed for positioning, navigation, and timing, can be used to retrieve the zenith tropospheric delay (ZTD) of the GNSS signal over an

observation station. The ZTD can be divided into zenith hydrostatic delay (ZHD) and zenith wet delay (ZWD). The ZHD can usually be obtained at a high accuracy from a standard empirical model together with some surface meteorological data at the station. The atmospheric water vapor information is contained in the GNSS-ZTD, more precisely, in the GNSS-ZWD, which can be converted into PWV. The GNSS were first applied to meteorological research in the 1990s (Bevis et al., 1992). Near real-time GNSS-ZTD products estimated from GNSS data processing have been routinely assimilated into numerical weather

models (NWM) for improving the performance of weather forecast (Bennitt and Jupp, 2012; Dousa and Vaclavovic, 2014; Guerova et al., 2016; Le Marshall et al., 2012, 2019). Research has demonstrated that the accuracy of GNSS-PWV can meet the accuracy requirements for most meteorological applications, and the applicability of GNSS-PWV for studying extreme weather conditions and climatic events has been investigated by many researchers (Bonafoni et al., 2013; Calori et al., 2016; Chen et al., 2018; Choy et al., 2013; Eugenia Bianchi et al., 2016; Shi et al., 2015; Wang et al., 2016, 2018; Zhang et al., 2015).

To obtain GNSS-PWV over a station, the first step is to estimate the ZTD of the station from GNSS data processing, and the two most common data processing strategies are the network approach and precise point positioning (PPP) approach (Ding et al., 2017; Douša et al., 2016; Guerova et al., 2016; Li et al., 2015; Lu et al., 2015; Rohm et al., 2014; Yuan et al., 2014; Zhou et al., 2020). The former uses double-differenced observations, while the latter uses un-differenced observations in the observation equation system. The ZWD can be obtained from subtracting the ZHD from the GNSS-ZTD, or directly estimated

if the ZHD has been corrected in the GNSS observation equation system, depending on the processing strategies adopted. Then the GNSS-PWV can be converted by:

$$PWV = \Pi \cdot ZWD \tag{1}$$

where $\Pi$ is the conversion factor (Askne and Nordius, 1987; Bevis et al., 1992), which is given by:

$$\Pi = \frac{10^6}{\rho_w R_v (\frac{k_3}{T_m} + k_2')} \tag{2}$$

where $\rho_w$ is the density of liquid water; $R_v$ is the specific gas constant for water vapor; $k_2'$ and $k_3$ are atmospheric refractivity constants; $T_m$ is the weighted mean temperature over the GNSS site, which is defined and approximated through



the following equation (Davis et al., 1985):

$$T_m = \frac{\int \frac{e}{T} dh}{\int \frac{e}{T^2} dh} \approx \frac{\sum_1^n \frac{\overline{e_i}}{\overline{T_i}} \Delta h_i}{\sum_1^n \frac{\overline{e_i}}{\overline{T_i}^2} \Delta h_i} \tag{3}$$

where $e$ and $T$ are the water vapor pressure (hPa) and absolute temperature (K) respectively; $n$ is the number of the layers; $\overline{e_i}$, $\overline{T_i}$ and $\Delta h_i$ are the mean water vapor pressure, mean temperature, and thickness of the $i$th layer respectively.

From Eq. (2), one can see that $T_m$ is a crucial variable for the determination of the conversion factor $\Pi$, which in turn affects the determination of PWV expressed by Eq. (1). The significance of obtaining accurate $T_m$ values has been demonstrated by previous researches (Bevis, 1994; Jiang et al., 2019; Ning et al., 2016; Wang et al., 2005, 2016). $T_m$ can be calculated from an observed atmospheric profile or NWM data (Wang et al., 2005, 2016). This observed atmospheric profile can be acquired from a radiosonde station, which is valid only for the sounding site. In fact, for GNSS stations, they are usually not co-located with any regional radiosonde stations, i.e. observed atmospheric profiles are unavailable, as a result, equation (3) is not applicable for GNSS stations. Moreover, even a GNSS station is co-located with a radiosonde station, due to the low temporal and spatial resolution of radiosonde data, the temporal resolution of its resultant $T_m$ is also low, which cannot meet the requirements of GNSS near real-time/real-time (NRT/RT) applications such as the conversion of GNSS-ZWD time series into PWV time series. Although atmospheric profiles can be obtained from NWM data, it is still difficult for users to obtain predicted results from the NWM data for NRT/RT GNSS-PWV sensing. Thus, it is of great importance to develop empirical $T_m$ models for time-critical applications. Some $T_m$ models have been developed with a focus of improving the accuracy of the $T_m$, and these empirical models can be classified into two categories. One category is such a model that depends on in situ surface temperature observation $T_s$, like the Bevis model, which is a simple linear function expressed as: $T_m = a + bT_s$ (Bevis et al., 1992). The two coefficients of such a linear function can be determined from the linear regression method based on long-term regional radiosonde data. However, the deployment of radiosonde stations is geographically sparse due to their high cost, and even worse is that there are no radiosonde stations at all in some areas. Yao et al. (2014a) developed a global latitude-dependent $T_m - T_s$ linear model using $T_m$ data from the global geodetic observing system (GGOS) and $T_s$ data from the European center for medium-range weather forecasts (ECMWF). Jiang developed a time-varying global gridded $T_m - T_s$ model using both $T_m$ and $T_s$ derived from ERA-Interim(Jiang et al., 2019). Ding (2018, 2020) developed two generations of global $T_m$ models using the neural network algorithm, in which temperature observations were required for the input and the models performed well.

The $T_m$ models mentioned above need in situ meteorological observations as the model's input. However, for GNSS stations, again, not all stations are equipped with meteorological sensors. In this case, the type of empirical models that are independent of meteorological observations had to be constructed. Yao et al. (2014b, 2012, 2013) used spherical harmonics to develop the GWMT, GTM-II and GTM-III models, in which both the height and the periodicity of $T_m$ were taken into account. Huang et al. (2019a) established a global $T_m$ model using the sliding window algorithm, which was based on varying latitude





and altitude. The widely used GPT2w model (Böhm et al., 2015) and its successor, GPT3 (Landskron and Böhm, 2018), provided gridded results with both 1°×1°and 5°×5° horizontal resolutions and the models also contain a few terms related to temporal variations in $T_m$ including the mean, annual and semiannual amplitudes. However, the height differences between the user site, e.g. a GNSS station, and its nearest four surrounding grid points were not considered. Recent studies have overcome this problem by providing $T_m$ values at various heights ranging from ground surface to the upper troposphere. He

et al. (2017) developed a voxel-based global model, named GWMT-D, using the $T_m$ values at four height levels of reanalysis data from the National Centers for Environmental Prediction (NCEP) to construct the voxels. The $T_m$ predicted for the user site can be obtained from an interpolation of the $T_m$ values at the eight grid points of the voxel that contains the user site. In recent studies, some researchers used a $T_m$ lapse-rate, the rate of change in $T_m$ with altitude, to correct the effect of the height element on $T_m$, e.g. IGPT2w (Huang et al., 2019b), GTm_R (Li et al., 2020) and GPT2wh (Yang et al., 2020). The GTrop

model (Sun et al., 2019), developed for predicting both ZTD and $T_m$, also took into account the $T_m$ lapse-rate, and it outperforms GPT2w obviously at altitudes under 10 km.

    As previously discussed, considering the lapse-rate in a $T_m$ model can improve the model's accuracy. However, the assumption that $T_m$ linearly varies with height, which many recently developed models were based on, may not agree well with the truth. In this research, a new global grid-based empirical $T_m$ model, named GGNTm, in which the vertical

nonlinear variation of $T_m$ was taken into account, was developed using a three-order polynomial function and ERA5 monthly mean reanalysis data over the 10-year period from 2008 to 2017, and the temporal variation in each of the four coefficients in the $T_m$ fitting function was also modelled with the variables of the mean, annual and semi-annual amplitudes of the 10-year time series coefficient.

    The outline of the paper is as follows. The features of the vertical nonlinear variation in $T_m$ were investigated in Sect.

2.2, then a three-order polynomial function fitting the 10-year $T_m$ profiles obtained from ERA-5 monthly mean reanalysis data was developed for the GGNTm model. In Sect.3, the performance of GGNTm was validated using the $T_m$ values from ERA5 hourly reanalysis and globally distributed radiosonde profiles in 2018 as the references. Conclusions are summarized in the final section.

## 2 Methodology for new model construction

### 2.1 Data Source

    ERA5 reanalysis data were the latest reanalysis data developed by the ECMWF. In this research, ERA5 monthly mean reanalysis data in the 10-year period from 2008 to 2017containing geopotential heights, temperatures, and specific humidity at 37 pressure levels with a horizontal resolution of 1°× 1° were downloaded from the web server of the Copernicus Climate Change Service (C3S, https://climate.copernicus.eu/climate-reanalysis). The geopotential heights, which are often used in

meteorology, were then converted to WGS-84 ellipsoidal heights. Water vapor pressure was calculated by (Nafisi et al.,



2012):

$$e = qp/(0.622 + 0.378q) \tag{4}$$

where $q$ is the specific humidity, which can be obtained from NWM data; $p$ is the atmospheric pressure.

## 2.2 Vertical variation of $T_m$

The ERA5 monthly mean products were used to analyze the vertical variation of $T_m$. As defined in Eq. (3), $T_m$ is a function of water vaper pressure and temperature. The variation in water vapor pressure in the vertical direction has been known nonlinear, while the vertical variation in temperature is often assumed to be a linear decay function (Dousa and Elias, 2014). In fact, there is such a phenomenon that temperature increases with the increase in height, the so-called temperature inversion, which occurs in both the upper atmosphere and near ground surface, meaning that the vertical variation in temperature is complex. As a result, $T_m$ in the vertical direction varies nonlinearly due to the irregular variations in both

water vapor pressure and temperature in the vertical direction. Fig. 1 shows four vertical profiles of water vapor pressure, temperature, and $T_m$ at the pressure levels that were under a 10 km ellipsoidal height at four grid points obtained from ERA5 monthly mean reanalysis in December 2017. It should be noted that the surface heights of the four grid points were different, and they were 0 m, 301 m, 13 m, and 180 m respectively. Sub-figures (a) and (b) show that, in the height range near the surface, temperature increases with the increase in height. In addition, all the four $T_m$ profiles (the black curves

with dots) in these sub-figures show a nonlinear variation trend. This implies that using a constant lapse-rate to model the vertical $T_m$ variation trend will result in large errors, i.e. the $T_m$ profiles cannot be accurately modelled through a constant $T_m$ lapse-rate. This finding aligns well with other researchers (e.g. (Yao et al., 2018)).

## 2.3 Three-order polynomial function for $T_m$ vertical fitting

A linear $T_m$ decay function with a constant $T_m$ lapse-rate can be expressed as:

$$T_m = \alpha + \beta(H - h_0) \tag{5}$$

where $\alpha$ is the $T_m$ value at the reference height $h_0$; $\beta$ is the $T_m$ lapse-rate and $H$ is the ellipsoidal height (km) of the user site. An equivalent expression of Eq. (5) is:

$$T_m = \alpha' + \beta'H \tag{6}$$

where $\alpha'$ denotes the $T_m$ value at 0 km ellipsoidal height. Some $T_m$ models were constructed based on this linear $T_m$ decay function. $T_m$ values from different height ranges can be used to calculate the $T_m$ lapse-rate. However, if $T_m$ varies nonlinearly in the vertical direction, the calculated $T_m$ lapse-rate values would have large errors. To overcome this problem,

in this research, a three-order polynomial function was selected for a new $T_m$ model:

$$T_m = a + bH + cH^2 + dH^3 \tag{7}$$

where $a, b, c, d$ are the four unknown coefficient parameters of the fitting function.

For the estimation of the two sets of unknown coefficient parameters expressed in equations (6) and (7), two schemes,



named scheme-1 and scheme-2, were for the two functions fitting the sample data of $T_m$ profiles of the 120 monthly mean reanalysis data over the 10-year period from 2008 to 2017 at each grid point. It should be noted that, only those $T_m$ values

from the heights under 10 km were selected for the sample data. For measuring how well the fitting function fits the sample data, the root mean square (RMS) of the differences between the $T_m$ values resulting from the fitting function and the sample data was calculated by:

$$RMS = \sqrt{\frac{1}{n}\sum_{i=1}^{n}\Delta_i^2}$$ (8)

where $\Delta_i$ is the residual of $T_m$ at the $i$th pressure level over the grid point. Fig. 2 shows the map for the mean of the RMSs of the fitting residuals of the $T_m$ from the aforementioned 120 monthly mean $T_m$ profiles (the samples) at each of the grid

points. The mean of the mean RMSs at all global grid points for scheme-1 and scheme-2 were 1.26 K and 0.30 K respectively. In addition, the RMS results in the left sub-figure (for linear function) were latitude-dependent, and small RMSs (blue) were in mid-latitude regions; large RMS values in both sub-figures were in Antarctica. Comparing the two subfigures, we could find that the RMS values shown in the right sub-figure were all very small and significantly smaller than that of the left sub-figure, meaning that the three-order polynomial fitting function superior to the linear fitting function.

**2.4 $T_m$ temporal fitting for new model**

In the previous section, the 10-year time series of coefficients in the three-order polynomial function expressed in Eq. (7) at each of the grid points were obtained from the least-squares estimation. Since they were not constant values, the temporal variation in each coefficient at each grid point needs to be further modelled for the new grid-based empirical $T_m$ model proposed in this study, GGNTm. The seasonal variation reflected in the 10-year time series of each of the coefficients $r =$

$a, b, c, d$ was analyzed using the fast Fourier transform (FFT), and results for seasonality and periodicity at point 60°N,120°E are shown in Fig. 3, which presented noticeable annual and semi-annual amplitudes. Similar periodicities were also found at other grid points. According to these characteristics, the fitting model for GGNTm containing three terms including mean, annual and semi-annual amplitudes for each coefficient time series at each grid point expressed by the following was adopted in this study:

$$r = A_0 + A_1 cos(\frac{doy-d_1}{365.25}2\pi) + A_2 cos(\frac{doy-d_2}{365.25}4\pi)$$ (9)

where $A_0$, $A_1$ and $A_2$ are the mean, annual and semi-annual amplitudes respectively; $doy$ denotes "day of year"; $d_1$ and $d_2$ are the initial phases of the annual and semi-annual periodicities, which are estimated together with the mean and amplitudes.

Then, the mean, annual and semi-annual amplitudes, and initial phases for each coefficient at each of the grid points over the globe (with the resolution of 1°× 1°) were determined using the least squares estimation method and the 10-year time series

of the coefficient. To calculate $T_m$ for a specific site and time, e.g. for a GNSS station at an observing time, the following 3-



step procedure needs to be carried out:

1) using Eq. (9) to calculate each of the four coefficients at each of the four grid points surrounding the user site;

2) using Eq. (7) to calculate the $T_m$ values at the height of the user site at each of the above four grid points (which is for the height dimension);

3) using an interpolation method, such as the inverse distance weighting or bilinear interpolation, on the four $T_m$ values from step 2) to obtain the $T_m$ value for the user site (which is for the horizontal dimension, as is shown in Fig. 4).

Till now the new model has been developed based on the 10-year sample data from 2008 to 2017. This model will be validated using the model predicted $T_m$ results in 2018 compared against the same year's (i.e. out-of-sample) reference data. Results will be discussed in the next section.

**3 Evaluation of GGNTm**

Our new model was developed for obtaining predicted Tm values over a site that is located in anywhere below the tropopause (assumed to be 10 km), which has significance for applications needing real-time or near real-time PWV converted from GNSS signals received by a GNSS receiver located in a flat area, an ocean area, a high mountainous area, or even a flight vehicle. For the performance assessment of the newly developed $T_m$ model, $T_m$ values over different pressure levels obtained from both ERA-5 reanalysis and radiosonde profiles in 2018 were selected as the references due to the fact that GNSS receivers may be located in any reasonable altitudes. The two statistics, bias and RMSE, were utilized to measure the systematic discrepancy and the accuracy of the model results. Their formulas are:

$$bias = \frac{1}{n}\sum_{i=1}^{n}(Tm_i^{model} - Tm_i^{ref}) \qquad (10)$$

$$RMSE = \sqrt{\frac{1}{n}\sum_{i=1}^{n}(Tm_i^{model} - Tm_i^{ref})^2} \qquad (11)$$

where $i$ is the index of the data element; $Tm_i^{model}$ denotes the model resultant $T_m$ value; $Tm_i^{ref}$ denotes the reference $T_m$ value; $n$ is the number of the $T_m$ values in the statistics.

**3.1 Comparison with ERA5 hourly data**

As the first set of the reference selected for the evaluation of the new model, ERA-5 hourly data (with the resolution of 5°× 5°) at 12:00 UTC on each day in 2018, which were out-of-sample data, were downloaded from the C3S. Then they were converted to $T_m$ profiles, and $T_m$ values at each of five pressure levels: 950hPa, 800hPa, 650hPa, 500hPa and 350hPa were used to calculate the bias and RMSE of the new model's $T_m$ results at the pressure level. In addition to the GGNTm model, other three empirical models developed in recent years including GPT3, GTrop and GWMT_D, in which different vertical



correction methods were also applied, were also evaluated for performance comparisons of GGTNm and these three models.

Table 1 shows the mean bias and mean RMSE of the $T_m$ values over all global grid points resulting from each of the above four models. As we can see, on a global scale, GGNTm outperformed all the other three models, especially at high pressure levels. The mean bias and RMSE of GPT3 varied significantly due to its lack of height refinement. GTrop was considerably better than GPT3, owing to its use of the $T_m$ lapse-rate, although its $T_m$ results were still had large errors at high pressure levels, which is most likely to be resulted from the neglecting of the nonlinear vertical variation in $T_m$. The large bias and RMSE of the GWMT_D results were possibly because its modelling was based on NCEP reanalysis data, and there may exist systematic differences between the reanalysis data from ECMWF and NCEP. However, GWMT_D still significantly outperformed GPT3, due to its voxel-based structure. Compared to GTrop and GWMT_D, GGNTm performed very well at all pressure levels. This is because the model accounted for the vertical nonlinear variation in $T_m$.

The results shown in Table 1 were the statistics of all global grid points at each of the five pressure levels selected. For more refined results, Fig. 5 shows the map for the RMSE of $T_m$ at each grid point at either the 950hPa or 500hPa pressure levels resulting from three models, and the reason for not selecting GPT3 here was due to its much poor performance. The 950hPa pressure level (the left column) results indicated that the RMSEs of $T_m$ resulting from all the three models were latitude-dependent and high accuracy $T_m$ values (in blue) were mainly in low-latitude belts. However, the results at the 500 hPa pressure level (the right column) indicated that the new model significantly outperformed the other two models. In addition, from the 950h pressure level results, the percentages of those RMSE values that were under 5 K from all the global grid points for GTrop, GWMT_D and GGNTm were 93.4%, 82.1% and 94.6% respectively; while the corresponding percentage values at the 500hPa level were 44.9%, 70.6% and 88.7%. These suggest that larger improvements made by the new model, i.e. GGNTm, over the other two models were at high-altitude pressure levels.

**3.2 Comparison with radiosonde data**

In this section, $T_m$ from radiosonde profiles were used as the reference for the performance assessment of the models selected. The original radiosonde data at all globally distributed stations in 2018 were downloaded from the website of the University of Wyoming (http://weather.uwyo.edu/upperair/). Different from the use of reanalysis data as the reference, water vapor pressure at each pressure level from a radiosonde profile was calculated through a mixing ratio:

$$e = Rp/(622 + R) \tag{12}$$

where $R$ denotes the mixing ratio (g/kg).

An additional data pre-processing procedure needs to be conducted for data quality control. Those poor radiosonde profiles needed to be identified and excluded from their use for the reference. The first check was for a valid mixing ratio value: if a pressure level lacks a valid mixing ratio value, then it is regarded invalid and thus to be excluded. After this initial checking was performed, further identifications were also carried out. A profile would be excluded if it met any one of the following four conditions:

(1)  the profile lacks surface meteorological observations;





(2)  the pressure value of the top pressure level is greater than 100 hPa;

(3)  the difference in the pressure values at two successive levels is under 200hPa;

(4)  the profile consists of a few pressure levels, e.g. if $\Delta P/n \leq 30\ hPa$ (where $\Delta P$ is the difference of the pressure values at the surface and the 100 hPa pressure levels, and $n$ is the number of all pressure levels from the surface to the 100hPa pressure levels), then the profile was regarded to have sufficient number of pressure levels, otherwise it would be excluded from the use in the testing.

Sounding balloons are commonly launched twice a day (at 00:00 and 12:00 UTC). In this research, only those stations
that had at least 300 profiles in 2018 were selected in the model performance assessment. After the above 5-step quality control procedure was performed, a total of 260140 profiles from 428 global radiosonde stations were finally used in the performance evaluation of four selected models.

Table 2 shows the mean bias and RMSE of surface $T_m$ values and $T_m$ values at all pressure levels from the surface to the 10 km height at all the aforementioned 428 global radiosonde stations resulting from each of the four models that were the
same as the ones tested in the previous section. For the surface $T_m$ results, the mean RMSE of GTrop and GGNTm were very close; GWMT_D was the worst, with the largest bias and RMSE values, which may be due to its low horizontal resolution (5°×5°). The other set of results, the RMSE of $T_m$ under 10 km, was calculated using the differences between model-predicted $T_m$ values and the reference Tm values over all pressure levels that with a height less than 10 km. A small RMSE of Tm under 10 km indicates that the model performs well at any altitudes below the tropopause. As we can see, GPT3 performed the worst
and significantly worse than the other three models, mainly due to lack of an appropriate modelling for the $T_m$ vertical variation; GWMT_D was slightly better than GTrop, possibly because the $T_m$ value from the former was interpolated from the $T_m$ values at four height levels; the mean bias of $T_m$ from the new model, GGNTm, was the least, with the value of −0.16, which was close to 0, meaning nearly unbiased; the RMSE of the new model was also the least, among the four models, which suggest that the vertical nonlinear variation of $T_m$ was modelled in the new model more accurately than the other existing
models.

Similar to Fig. 5, Fig.6 shows the map for the RMSE of $T_m$ values at each of the 428 radiosonde stations in 2018 at the surface pressure level (the left column) and all pressure levels that with a height less than 10 km (the right column) resulting from GTrop, GWMT_D, and GGNTm. It can be found that the RMSEs of all models were latitude-dependent, and those stations that had a large RMSE value were most located in north Africa and north-east America. At the four stations located in
Antarctic, their surface $T_m$ values were accurately modelled by these models. However, in terms of the RMSE of all pressure levels under 10 km, the GTrop results were relatively large at the four stations, whilst both GWMT_D and GGNTm performed well at three of the stations.

To further evaluate the performance of the three models at different height ranges under 10 km, the models' $T_m$ values from the aforementioned radiosonde profiles at the 428 global stations were divided into five height ranges, and Fig. 7 shows
each height range's bias and RMSE. We can see the following results 1) in the height ranges above 4 km, the GTrop results





had the largest bias and largest RMSE, and GWMT_D was considerably better than GTrop; 2) in low height ranges the GWMT_D results were the worst; 3) in all height ranges the GGNTm results were nearly unbiased and their accuracy varied little with height. The GGNTm model's consistent high accuracy in all height ranges suggest that the characteristics of the vertical nonlinear variation in $T_m$ is modelled by the proposed model more accurately than the other models.

### 3.3 Impact of GGNTm on PWV

The accuracy of GNSS-PWV over a GNSS site at an observing time is dependent upon the accuracies of the ZWD and the conversion factor. Uncertainty analysis has been conducted by some researches to study the uncertainty of the GNSS-derived PWV, including the uncertainty in the conversion factor (Bevis, 1994; Jiang et al., 2019; Ning et al., 2016). This research mainly focuses on the impact of the newly developed $T_m$ model on PWV, however, it is difficult to evaluate the impact of $T_m$ on the GNSS-PWV directly, as GNSS stations and radiosonde stations are usually not collocated. Thus, a theoretical model was adopted by several studies to study the error in PWV resulting from $T_m$ (He et al., 2017; Huang et al., 2019a; Li et al., 2020; Wang et al., 2005, 2016). The relationship of the relative error between $T_m$ and PWV can be expressed as:

$$\frac{\Delta PWV}{PWV} = \frac{\Delta \Pi}{\Pi} = \frac{1}{(1 + \frac{k_2'}{k_3} T_m)} * \frac{\Delta T_m}{T_m} \tag{13}$$

where $\Delta PWV$ is the absolute error in PWV resulting from $T_m$; $\Delta PWV/PWV$ is the relative error of PWV. Due to the fact that $k_2'/k_3$ is quite small ($\approx 6e^{-5}$ $K^{-1}$), the relative error of PWV resulting from $T_m$ is approximately equal to that of $T_m$. In this research, surface $T_m$ and PWV were obtained from the abovementioned radiosonde profiles downloaded from the University of Wyoming. The mean absolute error and mean relative error of PWV resulting from GGNTm model at the above-mentioned radiosonde stations are shown in Fig. 8. As is shown in the figure, the distribution of both mean absolute error and mean relative error of PWV resulting from GGNTm are latitude-dependent, and stations at high latitudes tended to have smaller absolute errors but larger relative errors compared to stations located in low-latitude regions. The mean of mean absolute error and mean relative error of PWV resulting from $T_m$ derived from GGNTm on surface level were 0.19 mm and 1.13%, respectively. $T_m$ values over different pressure levels of radiosonde profiles at the 428 radiosonde stations in 2018 were then divided into five height ranges to calculate the relative error of PWV resulting from selected empirical models. As is shown in Table 3, significant improvement can be obtained compared to the widely used GPT3 model, and the relative error of GGNTm varied little with height.

### 4 Conclusions

In GNSS meteorology, $T_m$ is an essential parameter for converting GNSS-ZWD to PWV over the GNSS observing station. In practice, the $T_m$ value over a GNSS station at an observing time is commonly obtained from an empirical $T_m$





model, such as GPT3, GTrop and GWMT_D. In this research, a new global gridded empirical $T_m$ model, named GGNTm,
was developed. In this model, the vertical nonlinear variation in $T_m$ was modelled using a three-order polynomial function
fitting ERA5 monthly mean reanalysis data over the 10-year period from 2008 to 2017; and seasonal variation terms, including
mean, annual and semi-annual amplitudes, for each of the coefficients in the polynomial function at each of global grid points
were also modelled based on the 10-year time series of the coefficient.

The performances of the newly developed GGNTm model was assessed and also compared with three existing models
GPT3, GTrop and GWMT using model predicted $T_m$ values in 2018 against two references in the same year: 1) $T_m$ from
ERA5 hourly reanalysis data and 2) $T_m$ from radiosonde profiles at 428 global radiosonde stations. Compared to the first
reference, the RMSEs of $T_m$ values resulting from GGNTm at five pressure levels over all the global grid points in 2018 were
significantly smaller than that of the other three models at high-altitude pressure levels. Compared to the second reference, the
mean bias and mean RMSE of $T_m$ resulting from GGNTm at all the 428 radiosonde stations in 2018 were $-0.34$K and 3.89K
respectively; and the mean bias and mean RMSE of $T_m$ resulting from GGNTm at all pressure levels from surface to 10 km
height were $-0.16$K and 4.20K respectively, which were significantly smaller than that of all the other three models. In all
five height ranges from surface to 10 km in altitude, the GGNTm results were nearly unbiased, and their accuracy varied little
with height. This result suggests that the characteristics of the vertical nonlinear variation in $T_m$ is modelled by the approach
proposed in this study more accurately than the existing models. In addition, the impact of GGNTm on GNSS-PWV was
analyzed using a theoretical function. The results showed that the relative error of PWV resulting from GGNTm outperformed
GPT3, GTrop and GWMT model.

The improvement in the accuracy of the new $T_m$ model has significance for GNSS meteorology. Our future work will
be focussing on using high temporal resolution atmospheric data such as ERA5 hourly reanalysis data, instead of monthly
mean data used in this study, to model the temporal variation of the coefficents in the $T_m$ fitting function for further improving
the accuracy of the GGNTm model.

**Data availability:**

ERA5 reanalysis: https://climate.copernicus.eu/climate-reanalysis
Radiosonde data: http://weather.uwyo.edu/upperair/

**Author contributions**

Peng Sun designed the experiments and wrote the original draft. Suqin Wu and Kefei Zhang reviewed and edited the manuscript.
Mofeng Wan and Ren Wang processed the ERA5 reanalysis data and radiosonde data.



## Competing interests:

The authors declare that they have no conflict of interest.

## Acknowledgements:

This work was funded by the National Natural Science Foundation of China (41730109, 41874040). The authors would also like to acknowledge the support of the Xuzhou Key Project (Grant No. KC19111) awarded in 2019 and the Jiangsu dual creative talents and Jiangsu dual creative teams program projects of Jiangsu Province, China, awarded in 2017. We would like to thank ECMWF and University of Wyoming for providing ERA5 reanalysis data and radiosonde profiles respectively.

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

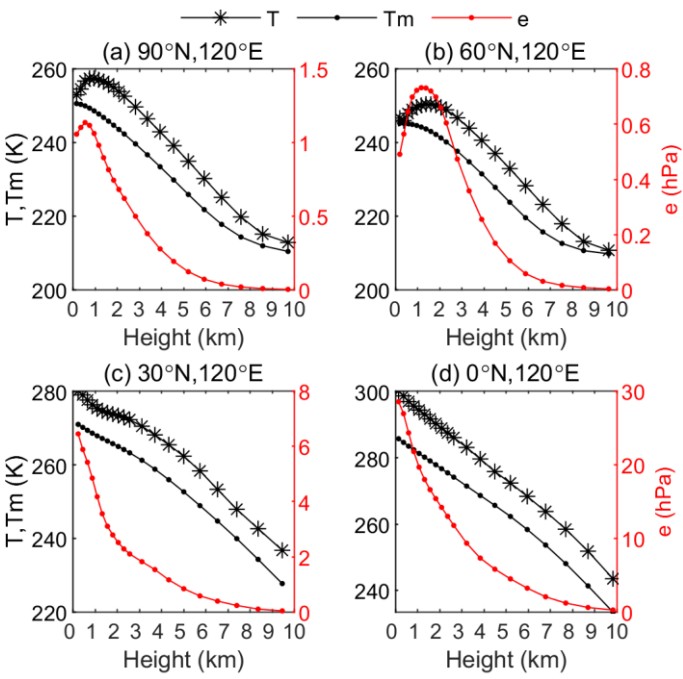

**Figure 1.** Temperature $T$, water vapor pressure $e$ and $T_m$ profiles obtained from ERA5 monthly mean reanalysis in December 2017 at 4 grid points: (a) 90°N, 120°E; (b) 60°N, 120°E; (c) 30°N, 120°E; (d) 0°N, 120°E.


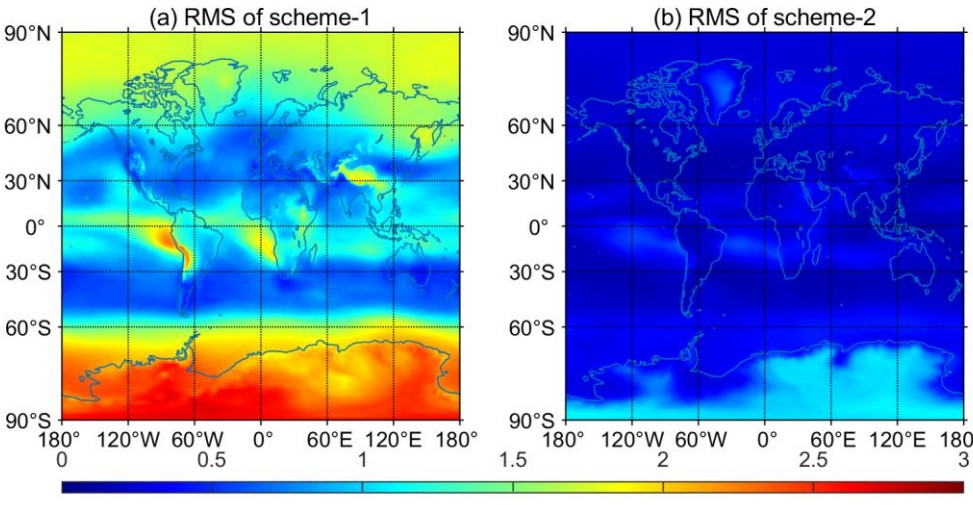

**Figure 2.** Mean of RMSs of the $T_m$ residuals of 120 monthly -mean profiles from the 10-year period at each grid point for scheme-1 (left, for linear function) and scheme-2 (right, for three-order polynomial function)




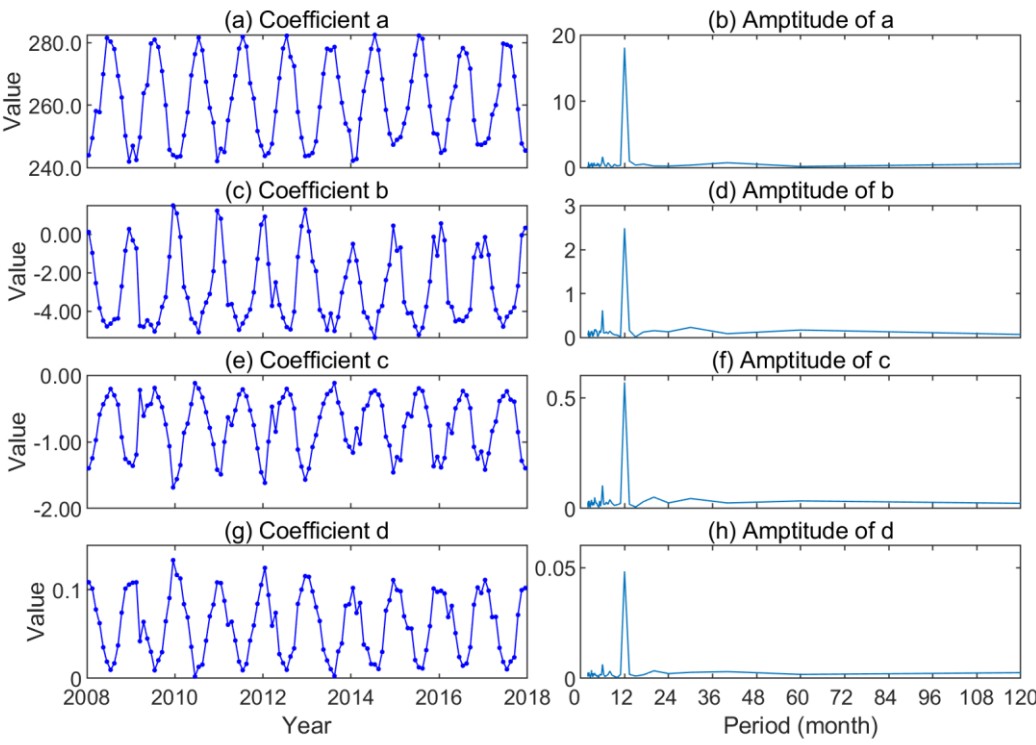

**Figure 3**. Periodicity reflected in the 10-year time series of each coefficient in the three-order polynomial function at 60°N,120°E.






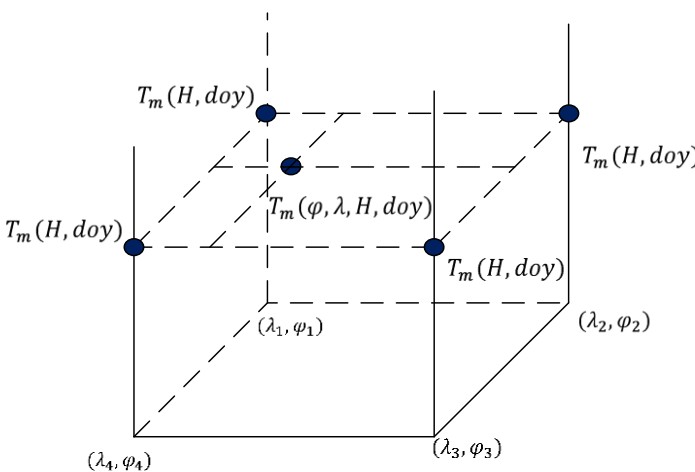

Figure 4. Spatial interpolation of the $T_m$ value for the target point $(\varphi, \lambda, H)$. After obtaining the $T_m$ values at the height $H$ at the four grid points (see the four grids on the top plane) by GGNTm model using Eq. (7), the $T_m$ value at the target point can be interpolated (the dashed rectangular).



**Figure 5.** RMSE of $T_m$ at each grid point at 950hPa (left column) and 500hPa (right column) pressure levels in 2018 resulting from GTrop, GWMT_D and GGNTm.


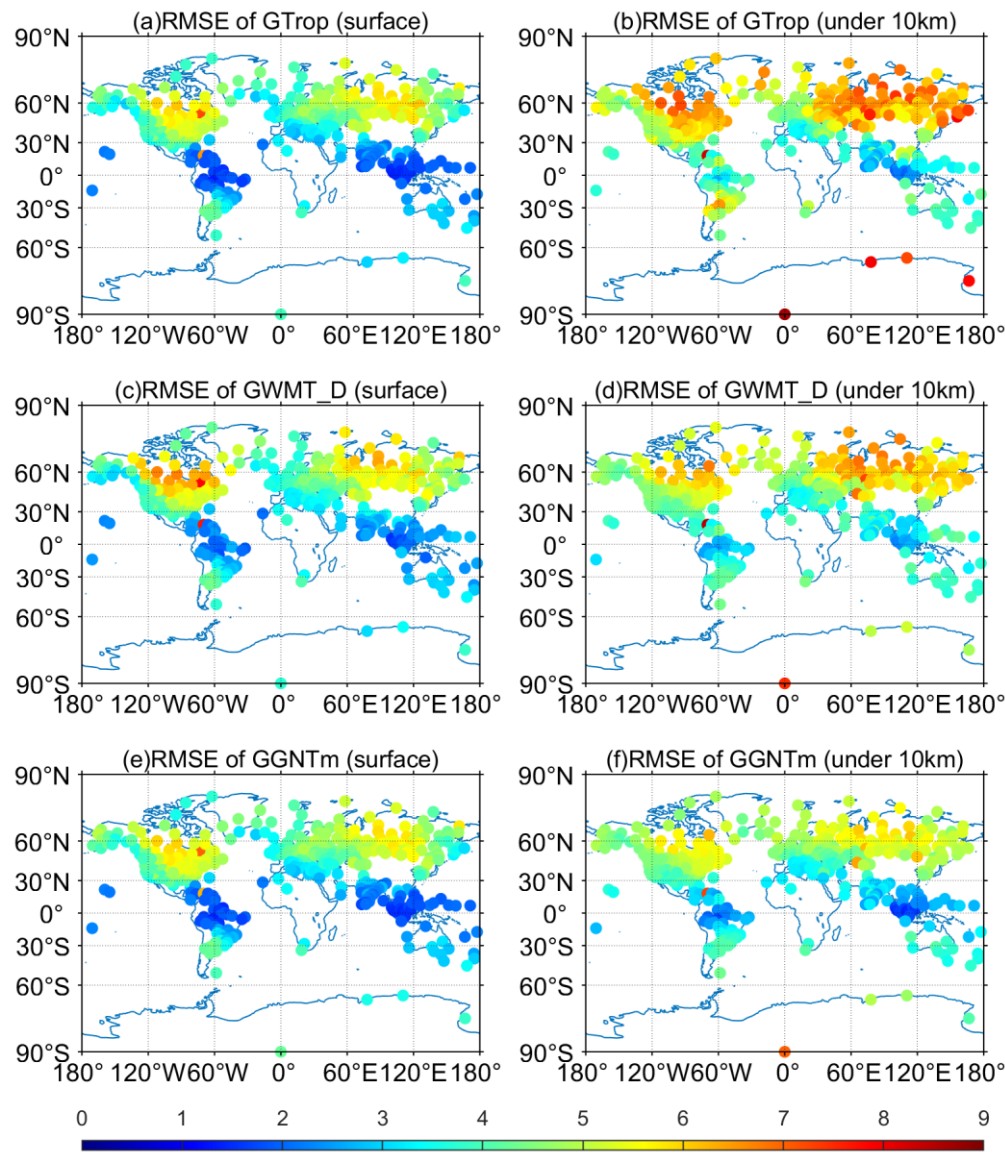

**Figure 6.** RMSE of $T_m$ at surface level (left column) and all pressure levels under 10 km (right column) at each of the 428 radiosonde stations in 2018 resulting from GTrop, GWMT_D, and GGNTm. The RMSE of Tm under 10 km was calculated using the differences between model-predicted Tm values and the Tm values over all pressure levels that with a height less than 10 km.





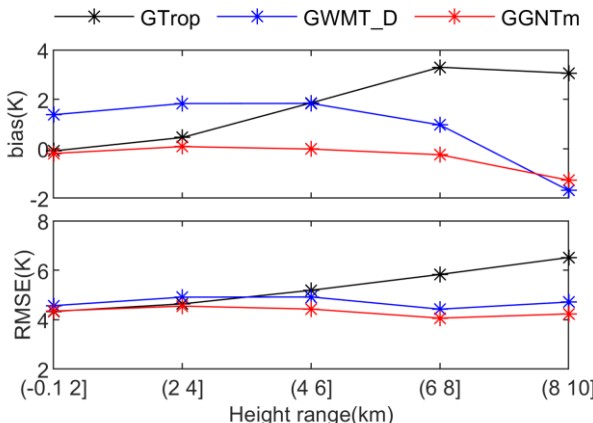

**Figure 7**. Bias and RMSE of $T_m$ from radiosonde profiles at 428 global radiosonde stations in each of five height ranges resulting from GTrop, GWMT_D and GGNTm .




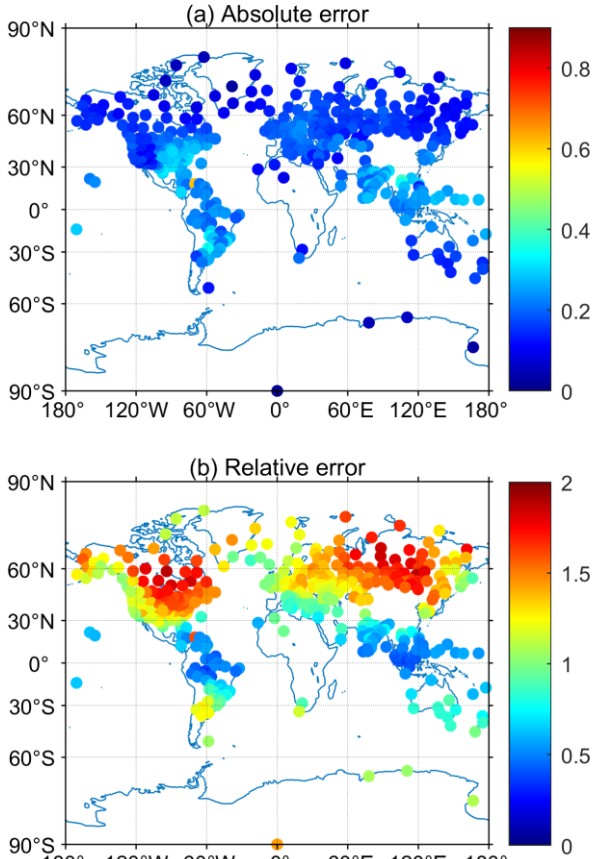

**Figure 8.** The mean absolute error and mean relative error of PWV at surface level resulting from $T_m$ derived from GGNTm at each of the 428 radiosonde stations in 2018




**Table 1.** Mean bias and mean RMSE of $T_m$ values at each of five pressure levels at UTC 12:00 at all global grid points in 2018 resulting from each of the four models selected.

| Pressure level (hPa) | Statistic (K) | Model | | | |
|---|---|---|---|---|---|
| | | GPT3 | GTrop | GWMT_D | GGNTm |
| 950 | | | | | |
| | mean bias | 0.74 | −0.14 | 1.68 | −0.43 |
| | mean RMSE | 4.25 | 3.39 | 3.98 | 3.35 |
| 800 | | | | | |
| | mean bias | 7.30 | −0.14 | 2.09 | 0.09 |
| | mean RMSE | 7.83 | 3.79 | 4.46 | 3.77 |
| 650 | | | | | |
| | mean bias | 15.91 | 0.76 | 1.84 | 0.15 |
| | mean RMSE | 16.68 | 4.14 | 4.58 | 4.07 |
| 500 | | | | | |
| | mean bias | 27.50 | 2.97 | 2.07 | 0.30 |
| | mean RMSE | 27.81 | 5.17 | 4.57 | 3.94 |
| 350 | | | | | |
| | mean bias | 42.27 | 5.71 | 1.90 | 0.78 |
| | mean RMSE | 42.47 | 7.12 | 3.93 | 3.02 |




**Table 2** Mean bias and mean RMSE of $T_m$ values at 428 globally distributed radiosonde stations in 2018 resulting from GPT3, GTrop, GWMT_D and GGNTm.

| Height | Model | Bias (K) | RMSE (K) |
|---|---|---|---|
| Surface | | | |
| | GPT3 | −0.36 [−7.87  5.81] | 3.97  [1.36  12.51] |
| | GTrop | 0.16 [−2.39  4.23] | 3.87  [1.35  7.22] |
| | GWMT_D | 1.30 [−1.74  5.64] | 4.07  [1.51  7.81] |
| | GGNTm | −0.34 [−3.17  3.74] | 3.89  [1.39  7.03] |
| Under 10km | | | |
| | GPT3 | 22.00  [6.78  27.29] | 27.67  [10.80 33.53] |
| | GTrop | 1.50  [−3.68  5.97] | 5.08  [1.90  8.68] |
| | GWMT_D | 1.16  [−0.20  6.18] | 4.61  [2.24  8.52] |
| | GGNTm | −0.16  [−3.81  4.69] | 4.20  [1.37  7.30] |

Note: the values within square brackets were the minimum and maximum.






**Table 2** Relative error of *PWV* in different height ranges.

| Height (km) | Relative error (%) | | | |
| --- | --- | --- | --- | --- |
| | GPT3 | GTrop | GWMT_D | GGNTm |
| (-0.1 2] | 1.83 | 1.23 | 1.31 | 1.23 |
| (2 4] | 5.32 | 1.38 | 1.47 | 1.35 |
| (4 6] | 10.32 | 1.60 | 1.55 | 1.38 |
| (6 8] | 15.56 | 1.91 | 1.46 | 1.31 |
| (8 10] | 20.59 | 2.25 | 1.47 | 1.29 |