# Peer review of "A new global grid-based weighted mean temperature model considering vertical nonlinear variation"

_Atmospheric Measurement Techniques, 2020_

## Referee Comment (RC1) · Maohua Ding (Referee) · 31 Oct 2020

General comments

The authors developed a weighted mean temperature (Tm) model called GGNTm. Similar to other recent published Tm models, GGNTm is a blind global Tm model and suitable for sites from the earth's surface to a height of 10 km. After reading this article for several times, my suggestion is that this article can be suitable for a possible publishing in Atmospheric Measurement Techniques (AMT). The following reasons prompted me to make a decision.

(1) A three-order polynomial function to model the vertical variation of weighted mean temperature (Tm) which is more advanced than a linear function. The coefficients in

the three-order polynomial function (i.e. a, b, c and d) were provided with global grids with a resolution of $1° \times 1°$ and annual and semi-annual variations. This modeling idea is relatively new.

(2)The state of the art meteorological reanalysis data (ECMWF ERA5 monthly mean reanalysis data) were used as modeling data of GGNTm.

(3) GGNTm has been compared with state of the art blind Tm models (i.e. GTrop and GWMT_D) and this new model show some improvements in accuracy.

Specific comments

Overall, this paper is well prepared. However, the authors should pay attention to the following problems.

(1)Although many latest studies [2][3][4] used the a linear function to describe the vertical variation of Tm, a nonlinear function has already been used by Yao et al. 2018[1]. Thus, it is not the first attempt using a nonlinear function. Although this reference is included in the reference list, I can not see any further discussions with their study. Their work has a very significant correlation with your study.

(2)It is good to compare GGNTm with GTrop and GWMT_D, since GTrop and GWMT_D stand for the state of the art blind Tm models. However, results of GPT3 are redundant and even meaningless. In fact, GPT3-Tm is GPT2w-Tm and many studies [1][2][3][4] have clearly pointed out the defect of GPT2w-Tm and the accuracy of GPT2w-Tm has been discussed for several times. I think just a few sentences can describe the defect of GPT3-Tm (GPT2w-Tm) and citing results of GPT2w-Tm in other references (e.g. reference [4]) is enough.

(3) I'm very curious that if the height of the GNSS user site is lower than the height of the grid points, will unpredictable results be produced?

Technical corrections

(1)Line 32: 'needing' can be replaced by 'requiring'.

(2)Line 40: 'a standard empirical model together with some surface meteorological data' can be replaced by 'the Saastamoinen-ZHD model together with measured site meteorological data'

(3)Lines119-120: The geopotential heights can not be convert directly to the ellipsoidal heights.

(4)Line 208: The statement 'there may exist systematic differences between the re-analysis data from ECMWF and NCEP' should be supported by some references.

(5)Line 281: 'surface Tm' can be replaced by 'site Tm'.

References

[1]Yao Y, Sun Z, Xu C, Xu X, Kong J (2018) Extending a model for water vapor sounding by ground-based GNSS in the vertical direction. J Atmos Sol Terr Phys 179:358-366

[2]Li, Q., Yuan, L., Chen, P. and Jiang, Z.: Global grid-based Tm model with vertical adjustment for GNSS precipitable water retrieval, GPS Solut., 24(3), 73.

[3]Yang F, Guo J, Meng X, Shi J, Zhang D, Zhao Y (2020) An improved weighted mean temperature (Tm) model based on GPT2w with Tm lapse rate. GPS Solut 24:46

[4]Sun Z, Zhang B, Yao Y. (2019) A Global Model for Estimating Tropospheric Delay and Weighted Mean Temperature Developed with Atmospheric Reanalysis Data from 1979 to 2017. Remote Sens 11(16):1893.

---

## Referee Comment (RC2) · Anonymous Referee #2 · 27 Nov 2020

Review of "A new global grid-based weighted mean temperature model considering vertical nonlinear variation" by Peng Sun et al.

General Comments:

The calculation of Tm is not a new issue. This work developed an empirical Tm model that takes the impact of altitude variation on Tm into account. Some interesting results were obtained. The manuscript is well written. However, I think some additional works are essential before considering the publication of this study.

Specific Comments:

We have seen from broader scientific publications that Tm was modeled by many authors by looking at the global or specific region or use the satellite techniques such as GNSS, GPS_RO, and climate reanalysis datasets. However, in my opinion, we should pay attention to some points

1. In the Introduction, you wrote that "not all GNSS stations are equipped with meteorological sensors." then you said the Tm models independent of meteorological observations "had to be constructed". Really? I don't think it is the only solution while there are some other methods to solve such a problem. For example, we can interpolate the measurements from nearby surface meteorological sensors to the GNSS stations followed by using the Ts-Tm model, or we can also interpolate the reanalysis vertical profiles over the sites. Right? So you should write more to convince me of the significance of your study.

2. What is the application area of your Tm model? For time-critical applications? Your Tm model is based on the ERA5 monthly mean reanalysis data. Theoretically, such monthly mean reanalysis data has no ability to capture the short-term variations of Tm. Furthermore, your Tm model is independent of real-time meteorological observations. Therefore, I am not sure about the ability of your Tm model for near-real-time applications. Maybe the error statistics of your Tm model is good. But these statistics indexes were also the "mean precision index" over a specific period. For near-real-time application, we should also pay attention to the short term performances of the Tm estimations, especially under some extreme weather conditions. I would like to see your discussions about these issues in detail. Giving some time series of Tm over some points may be helpful.

3. Or you can use your Tm model for climate research. Unfortunately, I didn't see any discussions about this. In fact, there are still some questions about climate application. What is the advantage of your model compared with other solutions, e.g. interpolation of reanalysis data? Are there enough GNSS observations located in "the ocean area, a high mountainous area, or even a flight vehicle" for demonstrating the advantages of your model in climate or weather issues?

4. I agree that Tm is "a crucial variable for the determination of the conversion factor II". However, the significance of II in determining GNSS PWV depends. Equation (13) in your study is not quite accurate. It may greatly exaggerate the impact of Tm errors on PWV calculations in many cases. Detailed discussions about the uncertainty budgets of GNSS PWV can be found in https://doi.org/10.5194/amt-9-79-2016 or https://doi.org/10.5194/amt-12-1233-2019. We can see that under some situations the barometric pressure observations may introduce much larger errors into the GNSS PWV estimations. So your serious discussions about the

improvement in GNSS PWV calculations brought by your Tm model will be grateful.

---

## Author Comment (AC1) · 28 Dec 2020

**Response to reviewer 1**

We are very grateful for the review work of the Reviewer #1 who has provided constructive comments.

We have examined all the comments and suggestions made carefully and relevant revisions have been made accordingly. The following are our responses and further explanations item-by-item:

*(1) Although many latest studies [2][3][4] used a linear function to describe the vertical variation of Tm, a nonlinear function has already been used by Yao et al. 2018[1]. Thus, it is not the first attempt using a nonlinear function. Although this reference is included in the reference list, I cannot see any further discussions with their study. Their work has a very significant correlation with your study.*

We agree with the comments made. This research is concentrated on developing a blind model that considering the nonlinear variation of Tm in the vertical direction and is independent of any other data sources. The nonlinear variation trend was found by Yao et al. (2018) and a nonlinear function integrating the linear function and the trigonometric function was proposed. However, a reference Tm at a specific height ($Tm_0$, $h_0$) that can be obtained from atmospheric profiles or other empirical models, is required as the input of the proposed model, which means that Tm cannot be determined by the model independently. Thus, it is not compared with our new model in the manuscript. Only three state of the art open-access blind models that can provide Tm directly were utilized in this research.

*(2) It is good to compare GGNTm with GTrop and GWMT_D, since GTrop and GWMT_D stand for the state-of-the-art blind Tm models. However, results of GPT3 are redundant and even meaningless. In fact, GPT3-Tm is GPT2w-Tm and many studies [1][2][3][4] have clearly pointed out the defect of GPT2w-Tm and the accuracy of GPT2w-Tm has been discussed for several times. I think just a few sentences can describe the defect of GPT3-Tm (GPT2w-Tm) and citing results of GPT2w-Tm in other references (e.g. reference [4]) is enough.*

We thank the reviewer for the comments about the inclusion of GPT3 in the comparison. We mostly agree with the reviewer to reduce the length of the discussion. Relevant revisions have been made to condense this part of the description (according to the reviewer's suggestions).

*(3) I`m very curious that if the height of the GNSS user site is lower than the height of the grid points, will unpredictable results be produced.*

Our new model is expressed as:
$$Tm = a + bH + cH^2 + dH^3$$
The first coefficient, $a$, is the empirical Tm value at the sea level at the grid point. Thus, the height of the grid point is set to zero, this means that the heights of most user sites are greater than the grid points. A radiosonde station that is located below the sea level ("Atyran" station, No. 35700) was also taken as the reference data for the evaluation of

the new model, and no obvious underperformance results were found (from GGNTm).

*(4) The geopotential heights cannot be converted directly to the ellipsoidal heights.*

Thanks for pointing this out. Yes, this is right. Although the geopotential heights cannot be converted to the ellipsoidal heights directly, an approximate conversion was conducted in this research. The equations given by Nafisi et.al. (2012) and Yilmaz (2008) were used for the conversion.

In addition, relevant revisions have been made in the revision in response to other technical corrections mentioned by the reviewer.

Finally, the reviewer is thanked again the careful review work and the constructive suggestions made.

**Reference:**

Landskron, D. and Böhm, J.: VMF3/GPT3: refined discrete and empirical troposphere mapping functions, J. Geod., 92(4), 349–360, doi:10.1007/s00190-017-1066-2, 2018.

Nafisi, V., Urquhart, L., Santos, M. C., Nievinski, F. G., Bohm, J., Wijaya, D. D., Schuh, H., Ardalan, A. A., Hobiger, T., Ichikawa, R., Zus, F., Wickert, J. and Gegout, P.: Comparison of Ray-Tracing Packages for Troposphere Delays, IEEE Trans. Geosci. Remote Sens., 50(2), 469–481, doi:10.1109/TGRS.2011.2160952, 2012.

Yao, Y., Sun, Z., Xu, C., Xu, X. and Kong, J.: Extending a model for water vapor sounding by ground-based GNSS in the vertical direction, J. Atmospheric Sol.-Terr. Phys., 179, 358–366, doi:10.1016/j.jastp.2018.08.016, 2018.

Yilmaz, N.: Comparison of different height systems, Geo-Spat. Inf. Sci., 11(3), 209–214, doi:10.1007/s11806-008-0074-z, 2008.

---

## Author Comment (AC2) · 28 Dec 2020

Response to reviewer 2

We would like to thank the reviewer for the constructive comments concerning our manuscript. We have examined all the comments carefully, and the following are our responses for the questions or comments:

(1) *In the Introduction, you wrote that "not all GNSS stations are equipped with meteorological sensors." then you said the Tm models independent of meteorological observations "had to be constructed". Really? I don't think it is the only solution while there are some other methods to solve such a problem. For example, we can interpolate the measurements from nearby surface meteorological sensors to the GNSS stations followed by using the Ts-Tm model, or we can also interpolate the reanalysis vertical profiles over the sites. Right? So you should write more to convince me of the significance of your study.*

This is a good question. As is correctly said by the reviewer, there are different ways to obtain Ts values, such as the interpolation method using the actual meteorological measurements nearby or the reanalysis data in the area. However, the performance of different methods varies due to the inherent nature of the individual method. This includes for example, the interpolation error due to different terrain elevations between the meteorological sensor's location and the point of interest in addition the interpolation methods used. As for the Bevis-like model (Ts-Tm) method, its fundamental assumption is the availability of local long-term radiosonde observations which may not be readily accessible for some regions. One of the possible ways to solve the availability issue is to use reanalysis data to to develop Ts-Tm models. However, such a reanalysis-based Ts-Tm model may not be as accurate as that derived from local radiosonde profiles. In addition, the timely availability is another concern. Another drawback in using the reanalysis data is its latency issue. For some time-critical applications, near real-time (NRT) / real-time (RT) Tm is essential for NRT/RT GNSS-PWV determination. All these form the basis for this research, the development of an empirical model that is independent of actual on-the-fly meteorological observations.

To clarify these points, relevant revisions have been made.

(2) *What is the application area of your Tm model? For time-critical applications? Your Tm model is based on the ERA5 monthly mean reanalysis data. Theoretically, such monthly mean reanalysis data has no ability to capture the short-term variations of Tm. Furthermore, your Tm model is independent of real-time meteorological observations. Therefore, I am not sure about the ability of your Tm model for near-real-time applications. Maybe the error statistics of your Tm model is good. But these statistics indexes were also the "mean precision index" over a specific period. For near-real-time application, we should also pay attention to the short-term performances of the Tm estimations, especially under some extreme weather conditions. I would like to see your discussions about these issues in detail. Giving some time series of Tm over some points may be helpful.*

Again, the reviewer has raised a very good question about the performance of our method since the monthly-mean data were used. In our research, ERA5 hourly reanalysis data at UTC 12:00 and globally distributed radiosonde profiles in 2018 were utilized to evaluate the performance of our new model. Both 24-hour and 12-hour variations of Tm have been used in the reference data for the evaluation of our new model in the form of "mean precision index". The performance of our model under extreme weather conditions has also been assessed (summer storm period in August and September 2018). The Tm values integrated from the radiosonde profiles at KingsPark radiosonde station (No.45005, Hong Kong) from August to September in 2018 were taken as the reference data. As is shown in Fig.1, the Tm values at the station predicted by our new model, as well as a Tm-Ts model (Tm=0.6195Ts+103.3452) developed using Tm and Ts series at KingsPark station (He et al., 2019) were compared against corresponding radiosonde measurements during the "summer storm" periods. The daily total rainfall data (published by Hong Kong Observatory, https://www.hko.gov.hk) during the two months are also shown in the figure. Heavy rainfall occurred frequently in Hong Kong during the two months, and a super typhoon, named "Mangkhut" landed near HongKong and caused torrential rain on 16[th] September. As is shown in the figure, our model shows clear outperformance during the two months. More experiments showed that the coefficients of a Ts-Tm models vary significantly with time (i.e. 0.6195 vs 0.58 for the linear part, 103.3452 vs 115.71 for the constant part, respectively), which means that a Tm-Ts model may have large errors during some periods.

[Figure]

Figure 1. Tm derived from radiosonde profiles, Ts-Tm model, GGNTm from August to September in 2018 at KingsPark station and the daily total rainfall at Hong Kong International Airport

*(3) Or you can use your Tm model for climate research. Unfortunately, I didn't see any discussions about this. In fact, there are still some questions about climate*

*application. What is the advantage of your model compared with other solutions, e.g. interpolation of reanalysis data? Are there enough GNSS observations located in "the ocean area, a high mountainous area, or even a flight vehicle" for demonstrating the advantages of your model in climate or weather issues?*

We agree that the reanalysis data are important and reliable products for climate research. Different from the reanalysis data, GNSS receivers are regarded as cost-effective equipment for meteorological research, the main advantage of the GNSS-based method is its real-time, stable, high-temporal-resolution and relative long-term capabilities. In fact, some preliminary research in relation to the long-term feature of the GNSS ZTD/PWV series and the relationship between GNSS-PWV and weather or climate issues have already been carried out (Bianchi et al., 2016; Bonafoni and Biondi, 2016; Calori et al., 2016; Chen et al., 2018; Choy et al., 2013; He et al., 2019; Junbo Shi et al., 2015, 2015, 2015; Rohm et al., 2014; Wang et al., 2018; Zhang et al., 2015).

As for the potential applicability of this research in ocean, mountain and flight-based etc. areas, We have noticed that some studies have extended the GNSS-PWV sensing to a shipborne GNSS receiver, or GNSS receiver that onboard other moving vehicles (Fan et al., 2016; Wang et al., 2019; Webb et al., 2016). Thus, we concentrated on developing a high-accuracy unbiased empirical model for predicting Tm values in any possible places, which is meaningful for GNSS meteorology.

*(4) I agree that Tm is "a crucial variable for the determination of the conversion factor II". However, the significance of II in determining GNSS PWV depends. Equation (13) in your study is not quite accurate. It may greatly exaggerate the impact of Tm errors on PWV calculations in many cases. Detailed discussions about the uncertainty budgets of GNSS PWV can be found in https://doi.org/10.5194/amt-9-79-2016 or https://doi.org/10.5194/amt-12-1233-2019. We can see that under some situations the barometric pressure observations may introduce much larger errors into the GNSS PWV estimations. So your serious discussions about the improvement in GNSS PWV calculations brought by your Tm model will be grateful.*

This is a good question. The revised paper has incorporated more discussions (see below) about the improvement of the GNSS -PWV brought by our new model.

We agree that the atmospheric pressure may introduce larger errors if the atmospheric pressure was observed with poor accuracy. However, we think that the errors in Tm should not be neglected in such conditions, as a large error in Tm could amplify the impact of the atmospheric pressure and hence may lead to more errors in the predicted GNSS-PWV. As for the improvement of the GNSS-PWV brought by GGNTm, a new experiment was conducted to study the impact of the errors in Tm on the GNSS-PWV using the ERA5 hourly reanalysis data that were utilized in Section 3.1. The ZWDs at each of the pressure levels over the globally distributed grid points (2664 grid points in total) were calculated through integration:

[revised manuscript text omitted]

---

## Referee Report (RR1)

Review of the revision of "A new global grid-based weighted mean temperature model considering vertical nonlinear variation" by Peng Sun et al.

To the Authors,

Thank you for considering some of my suggestions. However, the manuscript needs some more revisions.

1)    In the response, you answered the question (1) in my first report. However, I cannot see enough revisions to your last version manuscript. You should describe the significance of your study more clearly in the manuscript, not just in the response to my comments.

2)    Again, you answered my question (2) only in the response instead of the manuscript. And I am not satisfied with your answers. What I am concerned about is the ability of your model to reflect the short-term variations of Tm. From figure 1 in your response, we can see that the time series of the Tm from your model is quite smooth. Moreover, there is an obvious systemic bias between the outputs of $Tm=0.6195Ts+103.3452$ and the radiosondes. The coefficients are constant in equation $Tm=0.6195Ts+103.3452$ so its performance varies with time, while your model is time-varying. I would like to see the comparisons between your model and the other time-varying Ts-Tm model. It doesn't matter which model performs better because the main advantage of your model is the "height-varying" compared with the Ts-Tm model. I just want you to evaluate your model more comprehensively.

3)    In the abstract of the last version manuscript, you wrote "These results suggest that … can significantly improve the accuracy of model-predicted Tm for a GNSS receiver … (assumed to be 10 km)". From figure 8, I think there are of course improvements, but not significant. And you evaluated the accuracies at pressure levels, so it is not proper to say "below the tropopause (assumed to be 10 km)". I found the same problems many times in your manuscript.

---

## Author Response (AR2)

Dear Editor,

Thank you very much for your suggestions concerning our manuscript. We have examined all the comments by Reviewer#2 again and made more revisions accordingly. Please check the revised part highlighted in yellow in the new version.

Comment (1)

The revisions are made from line 71 to 75, 81 to 82, and 87 to 92.

Comment (2)

The revisions are made from line 194 to 197, 246 to 247, 327 to 328 and section 3.3. Two figures are shown in page 22 and 25.

Comment (3)

The revisions are made from line 41 to 47 and 107 to 110.

Comment (4)

The revisions are made in section 3.4.

We thank you very much for your time and consideration of this new manuscript.

Yours sincerely,
Peng Sun